



# Multiscale analysis of surface roughness for the improvement of natural hazard modelling

Natalie Brožová[1, 2]★, Tommaso Baggio[3]★, Vincenzo D'Agostino[3], Yves Bühler[1], Peter Bebi[1]

[1] WSL Institute for Snow and Avalanche Research SLF, Flüelastrasse 11, 7260 Davos Dorf, Switzerland
[2] Department of Environmental Systems Science, Swiss Federal Institute of Technology (ETH Zurich), Zurich, Switzerland
[3] Department of Land, Environment, Agriculture and Forestry, University of Padova, via dell'Università 16, 35020 Legnaro, PD, Italy

★ These authors contributed equally to this work.

*Correspondence to:* Natalie Brožová (natalie.brozova@slf.ch)

**Abstract.** Surface roughness influences the release of avalanches and the dynamics of rockfall, avalanches and debris flow, but is often not objectively implemented in natural hazard modelling. For two study areas, a treeline ecotone and a windthrow disturbed forest landscape of the European Alps, we tested seven roughness algorithms using a digital surface models (DSM)

with different resolutions (0.1, 0.5 and 1 m) and different moving window areas (9 m$^{-2}$, 25 m$^{-2}$ and 49 m$^{-2}$). The *vector ruggedness measure* roughness algorithm performed best overall in distinguishing between roughness categories relevant for natural hazard modelling (including shrub forest, high forest, windthrow, snow and rocky land-cover). The results with 1 m resolution were found to be suitable to distinguish between the roughness categories of interest, and the performance did not increase with higher resolution. In order to improve the roughness calculation along the hazard flow direction, we tested a

directional roughness approach that improved the reliability of the surface roughness computation in channelized paths. We simulated avalanches on a different elevation models to observe a potential influence of a DSM and a digital terrain model (DTM). Accounting for surface roughness based on a DSM instead of a DTM resulted not only in clearly higher roughness values of forest and shrub vegetation, but also in longer simulated avalanche runouts by 16–27% in the two study areas. We conclude that directional roughness is promising for achieving better assessments of terrain topography in alpine landscapes

and that applying an approach using DSM-based surface roughness could improve natural hazard modelling.

**Keywords.** digital terrain analysis, natural hazard, surface roughness, treeline ecotone

## 1 Introduction

Surface roughness is a topographic parameter commonly used to identify and characterize surface features, such as different

vegetation types (Stambaugh and Guyette, 2008) and geomorphological characteristics (Cavalli et al., 2008; McKean and Roering, 2004; Nguyen et al., 2005). Quantifying surface roughness is thus central for the estimation of various biophysical characteristics and ecosystem services (Grohmann et al., 2011; Koponen et al., 2004; Munsamy, 2017; Shepard et al., 2001;



Wiernga, 1993; Wu et al., 2018). With the increasing availability of high-resolution remote sensing data, it is increasingly possible to quantify surface roughness over larger areas and to estimate how related ecosystem services and climate feedbacks

change over time (Mina et al., 2017; Myers-Smith et al., 2015; Nel et al., 2014; Palomo, 2017). Surface roughness has effects on one of the most relevant ecosystem services in mountain regions: gravity-driven natural hazards. In particular, the occurrence and runout distance of rockfall, debris flows and snow avalanches are influenced by terrain roughness and land cover (Baroni et al., 2007; May, 2002; Michelini et al., 2017; Teich et al., 2014). In the following sections, we describe the most frequent gravity-driven natural hazards affecting the European Alps, highlighting why it is important to consider surface

roughness and land cover when modelling and predicting such phenomena.

Debris flows can be defined as gravity-driven flows consisting of interacting phases, mainly a debris and a fluid phase (Jakob et al., 2005; Pudasaini, 2012; Takahashi, 2000). Approaches to modelling flow propagation are numerous (Armanini et al., 2009; Frank et al., 2017; Hutter et al., 1996; O'Brien et al., 1993; Pudasaini and Mergili, 2019). However, a relatively small number of them consider the presence or absence of forest and the surface roughness (May, 2002). Ishikawa et al. (2000)

emphasize the importance of the land cover (especially forests) as an active prevention measure for stabilizing slopes and reducing the debris flow runout distance. Tree and shrub parameters are known to influence the velocity and runout distance in different parts of a debris flow fan (Ishikawa et al., 2000; Michelini, 2016). On the other hand, intrinsic physical characteristics and the solid volume concentration of the routing flow are important parameters in determining the interaction between debris flows and surface roughness. In particular, debris flows with a high concentration of the solid component

exhibit a strong interaction with forest structure (Michelini et al., 2017). A spatially distributed surface roughness map can increase the reliability of debris flow simulations. This aspect is of particular importance in extreme scenarios where the mass flow can spread outside the main channel path, propagating on other surface types.

Rockfall processes are influenced by topographic parameters (slope and terrain curvature), surface roughness and land cover (Pfeiffer and Bowen, 1989; Wang and Lee, 2010). Surface roughness and land cover influence the contact angle between the

rock and the surface, changing the velocity by rolling and sliding (Wang and Lee, 2010) and influencing the runout distance (Caviezel et al., 2019; Dorren et al., 2005; Lopez-Saez et al., 2016). Vegetation decreases the energy of moving rocks and eventually stops them (Jonsson, 2007). Tree density and size are fundamental characteristics for assessing the protection function of the forest (Dorren et al., 2015).

Surface roughness is an important parameter in relation to snow distribution (Lehning et al., 2011), and it is particularly crucial

in preventing weak layers and avalanche formation and release (Schweizer et al., 2003; Viglietti et al., 2010). The supporting force of tree stems and the heterogeneity of the forest snowpack, influenced by crown interception, reduce the release of slab avalanches (Bebi et al., 2009; McClung and Schaerer, 2006; Schneebeli and Bebi, 2004; Teich et al., 2012b). The anchoring effect of the vegetation in snow gliding has been demonstrated in several studies, and the density, height and heterogeneity of vegetation cover are crucial characteristics (Endo, 1983; Feistl et al., 2014; Höller, 2001, 2013). Furthermore, surface

roughness has a critical impact on the flow path and runout distance of avalanches (Bühler et al., 2011).



Terrain roughness is increasingly considered an important factor when evaluating vegetation effects on natural hazards and also more generally in large-scale hazard mapping. Moreover, vegetation effects on snow avalanches, rockfall and debris flows are often strongly dependent on the type of vegetation and on potential changes in vegetation over time (Bigot et al., 2009; May, 2002). Digital surface models (DSMs) capture surface characteristics and, depending on the frequency of acquisition,

detect land cover changes over time. Distinguishing among different vegetation types and assessing their effects on natural hazards is particularly important for spatially and temporally changing vegetation patterns in mountainous terrain. While the consideration of dense forest cover in natural hazard models is already advanced (Bühler et al., 2018; Feistl et al., 2015), this is clearly not the case for shrub forests, very open forest structures, and early successional stages of forest cover, which occur predominantly near treeline or after natural or anthropogenic disturbances (windthrows, bark beetle outbreaks, wildfires,

logging operations). Furthermore, treeline ecotones are generally shifting upwards and natural disturbances are expected to increase in the future, both due to global changes (Harsch et al., 2009; Seidl et al., 2017). Such regions are typical release and transition areas for gravitational hazards like snow avalanches, rockfall, landslides and debris flows. Widespread changes in landscape lead to shifts in vegetation composition (Tasser and Tappeiner, 2002), thus influencing surface roughness. It is necessary to understand which natural hazard processes can be expected with further changes and to map where these natural

hazards may occur, as the frequency intensity and extent of natural hazards may increase with decreasing surface roughness. Groups of trees and shrubs in treeline ecotones are not usually characterized as forest, even if they influence the release and dynamics of natural hazards (Elliott, 2017). It would thus be useful to improve the characterization of surface roughness calculated outside and inside mapped forest vegetation and to include lower vegetation, shrub forests and dead wood, which are not classified as forest. Natural disturbances, such as windthrow and bark beetle outbreaks, alter the forest structure and

thus change the forest protective function. Such natural disturbances are expected to become more frequent and severe under climate change (Bebi et al., 2017; Seidl et al., 2017), and forest protective functions may be reduced. The protective functions against snow avalanches, rockfall and debris flows are particularly at risk when a large-scale disturbance occurs and affects forests at the stand level. Windthrow creates a high degree surface roughness from downed trees, root plates and stumps. In the case of snow avalanches, surface roughness modifies snowpack properties and offers direct support (Schneebeli and Bebi,

2004), which, similarly to forest, may have the ability to hinder the formation of continuous weak layers (Schweizer et al., 2003). Leaving dead wood in place in protection forests after a windthrow event or bark beetle outbreak may thus offer sufficient protection capacity against snow avalanches until the post disturbance vegetation can take over this function (Wohlgemuth et al., 2017). Likewise, increased surface roughness from dead wood may considerably decrease the runout distance of rockfall processes (Fuhr et al., 2015; Bourrier et al., 2012; Ringenbach et al., 2021)

There are many algorithms quantifying surface roughness, indicating the variability of a certain topographic variable (slope, elevation, aspect, curvature and vector dispersion) within a certain area defined by a certain number of neighbouring cells (moving window) (Evans, 1984; Haneberg et al., 2005; Hobson, 1967; Philip and Watson, 1986; Sappington et al., 2007; Smith, 2014). In this study we consider roughness algorithms requiring a digital elevation model (DEM) as input. Surface roughness maps based on the analysis of a DEM are influenced by its resolution (Shepard et al., 2001) and moving window



size (Grohmann et al., 2011). Higher DEM resolutions (< 1 m) allow us to see more detailed terrain, but they are usually only available for smaller areas. DEM-based surface roughness algorithms calculate the roughness value by analysing a certain number of neighbourhood cells. However, the investigated natural hazards have a predominant diffusion direction identified as the combination of terrain slope and curvature. Some attempts to calculate the roughness along a given direction have been made, but they have not yet been applied to large-scale hazard mapping (Michelini, 2016; Trevisani and Cavalli, 2016).

In this study we compare the efficiency of seven widely used algorithms applied to high-resolution remote sensing data in distinguishing among different surface roughness categories in two study areas. We specifically addressed the following research questions: (1) How well can different surface roughness categories be distinguished with the selected algorithms? (2) What is the influence of the DSM resolution and moving window area on the roughness classification? (3) Is it possible to improve the roughness calculation by introducing a directional roughness along the predominant mass flow direction? (4) How much can a mass flow simulation improve if roughness is properly taken in account?

## 2 Methods

We identified and tested seven algorithms calculating surface roughness in order to understand which algorithm is the most suitable for terrain feature classification. The algorithms were chosen based on a literature review. Only those algorithms that are recognized for their ability to provide a accurate estimation of vegetation cover were selected. We tested these algorithms on two study areas to evaluate their performance in identifying the ground features of interest: biomass on the ground (disturbed forest), rocky surface, short vegetation and forest. We selected algorithms that use a DEM and we used the digital surface model (DSM), as it represents all the ground features of interest.

We selected two study areas, where a high-resolution DSM (0.1 m), derived by photogrammetry, and high resolution (0.5 m) LiDAR data were available and where relevant terrain features of disturbed mountain forest landscapes and treeline ecosystems were represented. To evaluate the effects of different cell resolutions, we tested three DSM resolutions, equal to 0.1, 0.5 and 1 m, resampling the 0.1 m photogrammetric DSM to 0.5 and 1 m (method: mean value). In previous studies the scale of the roughness calculation has been represented as a moving window identified by the number of cells (Grohmann and Riccomini, 2009; Michelini et al., 2017), which can result in different analysed areas (a moving window of $3 \times 3$ m with a resolution of 1 m results in an area of 9 m$^{-2}$, but with a resolution of 2 m the area is 36 m$^{-2}$). We therefore compared the roughness algorithms (Table 1) using different moving window areas instead of different moving windows (number of cells used). With this moving window area approach, the number of cells differs according to the DSM resolution, but the analysed area remains the same. The effect of scale was analysed using the smallest moving window areas in order to preserve the detailed terrain from the high-resolution DSMs. The moving window for different resolutions was approximated to the greater odd number. Using seven different algorithms to calculate roughness with three resolutions (0.1, 0.5 and 1 m) and three moving window areas ($3 \times 3$ m, $5 \times 5$ m and $7 \times 7$ m) resulted in a total of 63 combinations. We statistically tested (Sect. 2.3) how well these algorithms, in



different combinations of spatial resolution and moving window area, can distinguish between the seven roughness categories presented in Table 2.

After choosing the best performing algorithm to distinguish between the categories, we tested the difference between using a DSM and a DTM as an input for calculating the surface roughness. We also simulated an avalanche on both the DSM and

DTM to observe the influence of the surface roughness on the avalanche runout distance. The surface roughness and its impact on the runout distance of an avalanche are demonstrated when using a DSM, whereas the surface roughness is filtered out when a DTM is used.

## 2.1 Study areas

The two selected study areas, Braema and Franza, are located in the central and eastern European Alps, respectively (Fig. 1,

a). Braema (Fig. 1, b) is an example of a treeline ecotone with treeline expansion. Franza (Fig. 1, c) was impacted by the 2018 storm Vaia and is an example of fully windthrown forest.

**Figure 1: (a) Locations of the study areas in the central and eastern European Alps (map source: ©OpenStreetMap, 2021). (b) Braema, located close to Davos (Grisons, Switzerland; orthophoto in the background (swisstopo, 2021) and orthophoto in the front**
**from the drone flight 2019) and (c) Franza, located in the Dolomites (Veneto, Italy; orthophoto from the drone flight 2019).**





### 2.1.1 Braema

The Braema study area is located in south-east Switzerland near Davos (canton Grisons). The elevation varies between 1550 and 2300 m a.s.l., and the aspect is mostly north-eastern. The upper part ranges in slope from 30 to 45° and is covered mainly by meadow and rocky terrain. The area is steeper between 2000 and 2200 m a.s.l. (40–45°), where it is defined by open terrain and sparse vegetation. The study area also includes four valley channels. They are wider and less delineated at higher elevations but become narrower with decreasing elevation. At the lower elevations the banks of these gullies are stabilized by shrub forest dominated by green alder (*Alnus viridis*). Timber forest occurs below 1900 m a.s.l. on a moderately steep (less than 30°) to very steep slope (up to 45°). The dominant tree species is Norway spruce (*Picea abies*) with admixed European larch (*Larix decidua*) at high elevations (around 1800–2000 m a.s.l.). Avalanche barriers are present in the upper part of Braema, which served as a reference element for surface roughness in this study (Sect. 3.1).

The area of almost 1 km$^{-2}$ was surveyed on 17 June 2019 using a senseFly eBee+ drone equipped with an RTK GNSS system for accurate georeferencing (better than 5 cm). The 404 photos were acquired with a SODA camera (focal length 10.6 mm, pixel size 2.4 µm) at a mean flight height of 148 m above ground and an overlap of 60% (across track) and 70% (along track), resulting in an average ground sampling distance (GSD) of 3 cm. This imagery was successively processed with the software Metashape (Agisoft LLC, St Petersburg, Russia), resulting in a pointcloud with an average density of 263 points m$^{-2}$. The point cloud was then processed into a DSM with a cell size of 0.1 m.

### 2.1.2 Franza

The Franza study area is located in the Dolomites, Italy, near the village of Livinallongo del Col di Lana (Veneto region). The elevation ranges between 1650 and 1950 m a.s.l., and the aspect is south-western. The area extends for 12 ha and includes the Ru de Andraz stream in the lower part. The area was strongly affected by the storm Vaia of 29 October 2018, which uprooted a large part of the forest stand. The fallen trees were left on the ground and the area was not involved in forest management, except for the forest road, which was cleared of biomass. The remaining forest is just 5–10% of the original forest cover. The central upper part of this area is covered by meadows and young open forests, which were not affected by the storm. The disturbed forest was dominated by Norway spruce (*Picea abies*) with admixed silver fir (*Abies alba*). European larch (*Larix decidua*) was the only tree species to survive the storm. The mean inclination of the area varies between 30 and 40°.

The area was surveyed using a Phantom 4 drone (DJI, Shenzhen, China) with ground control points for image georeferencing. The drone flight took place on 26 October 2019 and the 971 images were successively processed in Metashape. The mean flight height was 45 m above ground and the image overlap was greater than 70%. The result was a DSM with a cell resolution of 0.05 m and a mean point density of 557 points m$^{-2}$ (ground control points residual error in x, y and z: 3.7 cm), which was resampled to 0.1 m (mean value method) and cropped to 11 ha for this study.





## 2.2 Surface roughness algorithms

In order to describe the roughness, which consists of both geomorphological features and vegetation, we selected and tested seven algorithms using a high-resolution DSM. We selected widely used algorithms and tested them with different spatial resolutions (0.1 m, 0.5 m and 1 m) and moving window areas (9 m$^{-2}$, 25 m$^{-2}$ and 49 m$^{-2}$) on both study areas. The selected
algorithms are summarized in Table 1.

**Table 1: Summary of the seven algorithms used to compute the terrain roughness.**

| Surface roughness algorithm | Abbreviation | Reference |
| --- | --- | --- |
| *Area ratio* | *AR* | Hobson, 1967 |
| *Vector ruggedness measure* | *VRM* | Sappington et al., 2007 |
| *Standard deviation of the profile curvature* | *SD_PC* | Grohmann et al., 2011 |
| *Standard deviation of the residual topography* | *SD_RT* | Grohmann et al., 2011 |
| *Standard deviation of the slope* | *SD_S* | Grohmann et al., 2011 |
| *Terrain ruggedness index* | *TRI* | Riley et al., 1999 |
| *Vector dispersion* | *VD* | Grohmann et al., 2011 |

### 2.2.1 Area ratio

The *area ratio* is the ratio between the real area and the flat area occupied by the square cell of the DSM (Hobson, 1967). The real area is computed using the slope algorithm implemented in GRASS GIS (Horn, 1981; Mitasova, 1985). The final map representing the area ratio is then smoothed using an average value within a moving window defined by the user. The *area ratio* is close to one for flat areas, while it extends up to an infinite value for extremely steep areas. In this study the algorithm was implemented as a shell script and run in the GRASS GIS environment (GRASS Development Team, 2021).

**2.2.2 Vector ruggedness measure**

For the *vector ruggedness measure*, the unit vector normal to the raster cell is decomposed in the relative x, y and z directions using the slope and the aspect of the cell through standard trigonometric functions (Durrant, 1996; Pincus, 1956). Its measure and computation are fully described in Sappington et al. (2007). The resultant vector is calculated over a user-defined moving window. The strength of the vector is normalized for the total number of cells included in the moving window. In this study
the algorithm was implemented as a raster module in GRASS GIS called r.vector.ruggedness.



### 2.2.3 Standard deviation of slope and profile curvature

The *standard deviation (SD) of the slope* represents the slope standard deviation within a defined moving window (Grohmann et al., 2011). The slope is derived from the DSM using the algorithm r.slope.aspect implemented in GRASS GIS derived from the formula proposed by Horn (1981). With the same approach, the *standard deviation of the profile curvature* (second
derivative of the elevation) is computed within a moving window (Grohmann et al., 2011). Here, both algorithms were implemented in a shell script and run in GRASS GIS.

### 2.2.4 Standard deviation of residual topography

The *SD of residual topography* is computed as the SD of the difference between a smoothed DEM and the original one. The SD is calculated within a moving window defined by the user. This approach is widely used because it can be applied to
different data types, such as point clouds (Vetter et al., 2012), satellite imagery (Gille et al., 2000; Schumann et al., 2007) and DEMs (Glenn et al., 2006; Trevisani and Cavalli, 2016). In this study we calculated the smoothed DEM as the average value within a moving window $10 \times 10$ m independently from the input DEM resolution. The moving window used to compute the smoothed surface is automatically adjusted according to the model resolution. In this study this approach was implemented in a shell script and run in GRASS GIS.

### 2.2.5 Terrain ruggedness index

The *terrain ruggedness index* (*TRI*) is calculated as the mean change in elevation between a centre cell and its neighbours defined by the user (Riley et al., 1999). It represents the absolute variation between the centre cell and the surrounding cells. The index is similar to the average deviation of the centre absolute value, but it differs by the use of the centre cell. In the *TRI*, the centre cell is used as the reference instead of the average value of the cells within the defined moving window, thus
emphasizing the roughness. For this reason, *TRI* is more effective for highlighting the terrain features, especially in a small-scale analysis. In this study the algorithm called r.tri was implemented as a raster module in GRASS GIS, where it is possible to define the size of the moving window.

### 2.2.6 Vector dispersion

*Vector dispersion* is calculated as the orientation of a three-dimensional surface for the region of interest (Hobson, 1967). The
different planes of the DSM are approximated by normal unit vectors and the relative mean, dispersion and strength are calculated using the methods explained by Fisher (1953) and successively adapted by McKean and Roering (2004). This algorithm measures the degree of dispersion of the unit vectors in a given moving window. Here, the script was implemented as a raster GRASS GIS module called r.roughness.vector (Grohmann et al., 2011). To obtain the vector strength, the direction cosines maps are first calculated (Eq. 1) and successively summed for a user-defined moving window (Eq. 2). The vector
strength (*R*) and vector dispersion (*k*) are derived with Eq. 3 and Eq. 4, respectively, where *N* is the number of vectors. The



*vector dispersion* has low values for regular smooth surfaces because the vectors are parallel and the vector strength becomes closer to the number of vectors. This algorithm is sensitive to small-scale variation in elevation and is therefore considered suitable for detecting vegetated areas.

$$x_i = sin\theta_i cos\phi_i \quad y_i = sin\theta_i sin\phi_i \quad z_i = cos\theta_i \tag{1}$$

$$\overline{x} = \sum_{i=1}^{N} x_i \quad \overline{y} = \sum_{i=1}^{N} y_i \quad \overline{z} = \sum_{i=1}^{N} z_i \tag{2}$$

$$R = \sqrt{\overline{x}^2 + \overline{y}^2 + \overline{z}^2} \tag{3}$$

$$k = (N - 1)/(N - R) \tag{4}$$

**2.3 Design and statistical analysis of roughness categories**

Seven different roughness categories (Table 2) were chosen using orthophotos from the two study areas (Fig. 2). In order to distinguish between the categories "very smooth" and "smooth", and between "shrub forest" and "high forest", we used a vegetation height model (VHM), which we calculated as the difference between the digital surface model and digital terrain model (VHM = DSM – DTM; we used the LiDAR data described in Sect. 2.5). The "snow" category was selected as the control, since in our case this surface is the smoothest and should therefore have the lowest roughness values. The category "very smooth" is dominated by features with heights up to 0.5 m, the category "smooth" mainly includes features from 0.5 to 1 m height, and both of these categories are dominated by lower vegetation, which is not considered important for the interaction with natural hazards. "Shrub forest" is mainly composed of green alder and smaller trees between 3 and 5 m tall, while the "high forest" category has a minimum tree height of 10 m.

Control areas of 10 × 10 m were manually selected using the orthophotos in order to extract the calculated roughness values and compare these values for different categories. The number of values extracted per category depended on the spatial resolution. A higher spatial resolution (0.1 m) results in 10,000 values per feature, a medium resolution (0.5 m) results in 400 values per feature, and a lower resolution (1 m) results in 100 values per feature. We randomly sampled all the values to obtain 1000 values per roughness category for analysis. We statistically analysed (paired Wilcoxon test) the algorithms to determine the overlapping distribution of pairs of roughness categories. The tested algorithm in the corresponding resolution and moving window was able to distinguish between the roughness categories in cases where there was a significant difference (p-value <0.05) between the categories. In order to obtain a classification based on threshold values for a technical purpose, we analysed the kernel density distribution between the roughness categories (Table 2) to determine the point of minimum overlap. We used the *overlap* function (*overlapping* package; Pastore, 2018; Pastore and Calcagnì, 2019) implemented in R (R Core Team, 2021). This intersection is the threshold between two roughness categories.





**Table 2: Roughness categories using orthophotos and the vegetation height model (VHM) were selected to evaluate the different surface roughness algorithms.**

| Roughness category | Braema | Franza |
|---|---|---|
| snow | x | |
| very smooth | x | x |
| smooth | x | x |
| shrub forest | x | |
| high forest | x | x |
| rocky | x | |
| windthrow | | x |

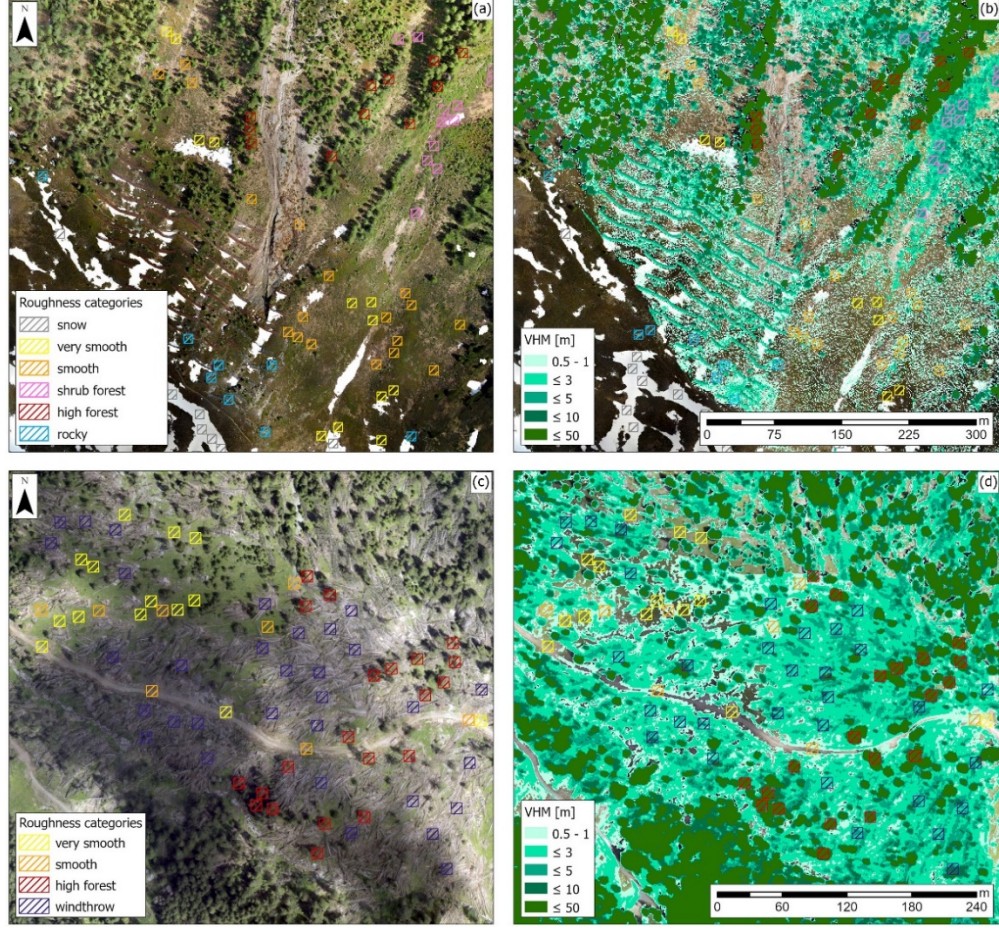

**Figure 2: Roughness categories were selected based on the orthophoto alone (a) and (c) (drone flights, 2019) and with the vegetation height model (VHM, produced as the difference between the DSM and DTM provided by the Federal Office of Topography for Braema (swisstopo, 2018) and Veneto region for Franza; (b) and (d)). The Braema study area (Grisons, Switzerland) is shown in (a) and (b), and the Franza study area (Veneto, Italy) is shown in (c) and (d).**





## 2.4 Directional roughness

Since mass flows have a propagation direction, one of our aims was to improve the roughness calculation along the expected direction of diffusion. For this purpose, we modified the *SD of residual topography* algorithm to test the roughness improvement along open slopes, valleys and gullies. The directional roughness is computed as the SD of the residual topography where only a subset of the neighbourhood cells are analysed for 16 directions. The roughness direction is identified using a manually designed polyline. In accordance with the direction given by the polylines, the algorithm computes the SD of six or four cells, without considering the central cell value of the moving window (the resolution used was 1 m and the cell moving window area was 9 $m^{-2}$). We calculated the directional roughness for the Braema study area only. In order to better understand the effects of the directional roughness, we manually identified four transects (Fig. 3) and compared the directional and non-directional roughness.

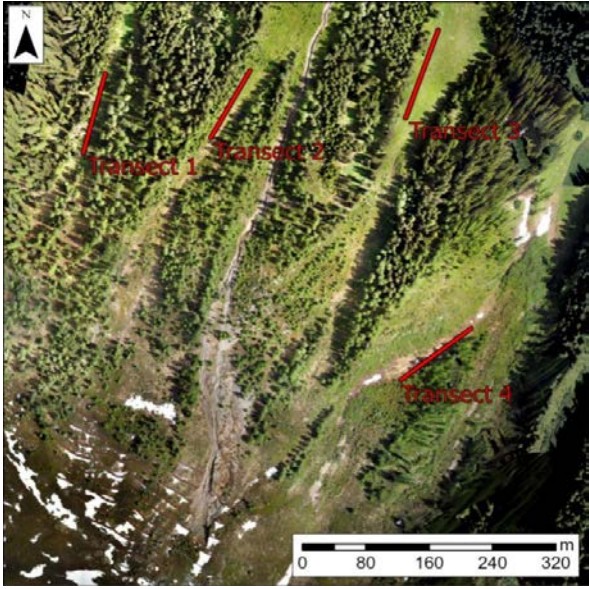

**Figure 3: Four transects within gullies in the Braema study area (Grisons, Switzerland; orthophoto from drone flight, 2019). The surface roughness was analysed using non-directional and directional *SD of residual topography*.**

## 2.5 LiDAR-based roughness

For the best performing algorithm, we compared the terrain roughness from the DSM and a LiDAR-based DTM. LiDAR data were acquired in July 2019 (> 35 points $m^{-2}$) for Franza and in August 2015 for Braema (> 18 points $m^{-2}$). Regarding the Franza study area, the DSM and DTM were already produced for the Veneto region as a raster layer at the final resolution of 0.5 m. The LiDAR survey of the Braema area was acquired with an LMS-Q 780 and was part of a larger surveying campaign and were provided by the Federal Office of Topography. The final DEM products were resampled to a resolution of 0.5 m.





Based on the results of the roughness algorithm evaluation, we calculated the terrain roughness for the DTM and the DSM using the *vector ruggedness measure* algorithm (moving window area 49 m$^{-2}$ and cell size 0.5 m). We then plotted the results to highlight the differences in terrain roughness.

**2.6 Case study: snow avalanche modelling**

To investigate the importance of terrain roughness on the numerical simulation results, we implemented a snow avalanche
simulation. We performed a total of four simulations using two types of terrain morphology (the LiDAR-derived DTM and the DSM) for both of the study areas. The simulation tool that we applied is the snow avalanche module of RAMMS (Christen et al., 2010), version 1.7.20. We identified one release area for each study area based on topographic and vegetation analysis (terrain slope, curvature, land cover). The release depth was homogeneous and we set it to 1 m, accounting for a total volume of 1457.9 m$^3$ (Braema) and 284.8 m$^3$ (Franza). We used the automatically calculated friction values for different topographic
conditions based on the return period (30 years) and volume (small and tiny for Braema and Franza, respectively), as described in the RAMMS user manual (Bartelt et al., 2017). The forested areas are based on the forest characteristics specified in Swiss law (Brändli and Speich, 2007) and delineated using an orthophoto. We determined the runout distance manually as the projected run length in the main flow of the avalanche, where the maximum flow depth of the simulated avalanche drops to zero (Brožová et al., 2020). We also evaluated the maximum flow height over the simulation duration.

**3 Results**

**3.1 Roughness classification and algorithm evaluation**

Four of the seven selected roughness algorithms were found to be suitable for distinguishing the investigated vegetation types and other land-cover categories, as shown in Fig. 4, without any overlapping pairs. However, there were important differences according to the spatial resolution and the moving window area considered for the analysis. With only 4 of 441 possible pairs
of overlapping distributions (red arrows in Fig. 4), the algorithms generally distinguished better between categories if applied using the lowest resolution of 1 m compared with applications using higher resolutions of 0.5 m (16 overlapping out of 441 pairs) and 0.1 m (9 of 441 pairs). In the lowest resolution, only the *area ratio*, *SD of slope* and *vector dispersion* algorithms showed overlapping distributions in some of the categorization and only the *vector dispersion* algorithm failed in two moving window sizes (9 m$^{-2}$ and 49 m$^{-2}$) in comparison to higher resolutions, where there was only one failure (Fig. 4). Overall, we
found the best differentiation between the roughness of different land-cover categories for the largest considered moving window area of 49 m$^{-2}$ in combination with the resolution of 1 m (no pairs of overlapping distribution) compared with other combinations of smaller moving window areas of 9 or 25 m$^{-2}$.

The best performing algorithm without any significantly overlapping distributions of pairs in all spatial resolutions (0.1 m, 0.5 m and 1 m) and all moving window areas (9 m$^{-2}$, 25 m$^{-2}$ and 49 m$^{-2}$) was *vector ruggedness measure*. Other algorithms that
performed well in distinguishing the roughness categories were *SD of profile curvature*, *SD of residual topography* and *SD of*



*slope*. *SD of slope* had one overlapping distribution of roughness values for the category shrub forest and high forest with a resolution of 1 m and a moving window area of 9 m$^{-2}$. *SD of residual topography* did not distinguish between very smooth and smooth when combined with a resolution of 0.1 m and a moving window area of 9 m$^{-2}$, or between shrub forest and windthrow (resolution of 0.5 m and moving window area of 25 m$^{-2}$). *SD of profile curvature* did not accurately differentiate between the

categories high forest and windthrow (resolution of 0.1 m and moving window area of 49 m$^{-2}$, and resolution of 0.5 m and moving window area of 25 m$^{-2}$) and between the categories very smooth and smooth (resolution of 0.5 and moving window area of 25 m$^{-2}$). The algorithms *vector dispersion* (4 pairs of overlapping distribution), *terrain ruggedness index* (6 pairs) and *area ratio* (13 pairs) were overall less efficient in distinguishing between different roughness and land-cover categories.

Surface roughness calculated with the seven different algorithms and normalized using the same colour range (Fig. 5 and Fig.

A1 in Appendix, for the Braema and Franza study areas) revealed important differences in the ability to identify specific terrain and vegetation types. As visible for the overall best performing combination of resolution and moving window (1 m and 49 m$^{-2}$) in Fig. 5, all algorithms distinguished accurately between high vegetation (forest) and other vegetation types. Nevertheless, some of the algorithms (*vector dispersion*, *SD of residual topography* and *area ratio*) failed to detect the avalanche barriers correctly and falsely identified them as rather smooth. Also, small gullies were not clearly separated with some of the

algorithms and were particularly poorly visible with the algorithms *SD of profile curvature* and *SD of slope*, whereas they were successfully identified with moderate roughness values by the other algorithms. Smooth surfaces were visualized with lower roughness values (darker blue in Fig. 5) by algorithms like *vector ruggedness measure*, *SD of residual topography* and *vector dispersion* [Fig. 5 (2, 4 and 7)]. Other algorithms [Fig. 5 (1, 3, 5 and 6)] assigned these smooth surfaces rather high roughness values (lighter blue to cyan blue in Fig. 5).

**Table 3: Thresholds of roughness values between roughness categories calculated using the *vector ruggedness measure* algorithm, with 1 m resolution and a moving window area of 49 m$^{-2}$.**

| Roughness category | Threshold value |
| --- | --- |
| snow to very smooth | 0.006 |
| very smooth to smooth | 0.017 |
| smooth to rocky | 0.037 |
| rocky to shrub forest | 0.089 |
| shrub forest to windthrow | 0.171 |
| windthrow to high forest | 0.301 |

The *vector ruggedness measure* algorithm showed the least overlapping of pairs and was found to be the best performing algorithm for our application. We determined the intersecting points within the densities of neighbouring roughness categories

(Table 3), which may be used as thresholds for surface classification based on roughness.



**Figure 4: Distribution of roughness values according to different roughness categories (1 – snow, 2 – very smooth, 3 – smooth, 4 – shrub forest, 5 – high forest, 6 – rocky, 7 – windthrow) for seven algorithms (*area ratio*, *vector ruggedness measure*, *SD of profile curvature*, *SD of residual topography*, *SD of slope*, *terrain ruggedness index* and *vector dispersion*) for the spatial resolution of 1 m. Red arrows show the overlapping distribution for a pair of categories that the given algorithm fails to distinguish.**


**Area ratio**

**Vector ruggedness measure**

**SD profile curvature**

**SD residual topography**

**SD slope**

**Terrain ruggedness index**

**Vector dispersion**

**Orthophoto**

**DSM hillshade**

Roughness

high

low

N

0   75   150   225   300   m

Spatial resolution: 1 m
Moving window: 7x7 m
MW area: 49 m$^{-2}$

**Figure 5: Calculated surface roughness in the study area Braema using the seven investigated algorithms:** *area ratio* **(1),** *vector ruggedness measure* **(2),** *SD of profile curvature* **(3),** *SD of residual topography* **(4),** *SD of slope* **(5),** *terrain ruggedness index* **(6) and** *vector dispersion* **(7). The same area is presented as an orthophoto (8) (drone flight, 2019) and in DSM hillshade (9). All algorithms were calculated based on the overall best performing combination of spatial resolution (1 m) and neighbourhood (moving window 7 × 7 m). To improve the visualization and compare the roughness maps, we normalized them with the 25th percentile as the minimum value and the 75th percentile as the maximum one.**




## 3.2 Directional roughness

The analysed surface roughness within gullies and valleys in the study area using the *SD residual topography* algorithm showed lower values for directional roughness than the non-directional one. The calculated roughness in the mass flow direction of propagation differed significantly from values calculated without using the direction ($p < 0.05$, Wilcoxon test; Fig. 6 and Fig. 7). In particular, for some transect parts the non-directional values were twice as large as for the directional ones. In other parts, the two roughness maps were almost equal. However, the non-directional roughness never exceeded values of the

directional roughness within the selected transects.

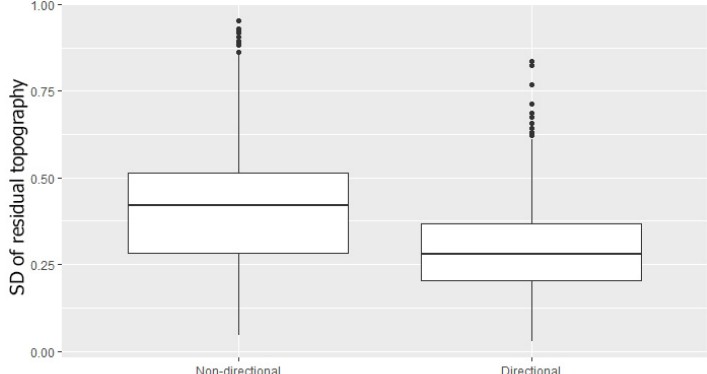

**Figure 6: Surface roughness values calculated using non-directional and directional *SD of residual topography*. Using direction in the calculation of the surface roughness within the gullies resulted in values significantly lower ($p < 0.05$) than those calculated with the non-directional method.**

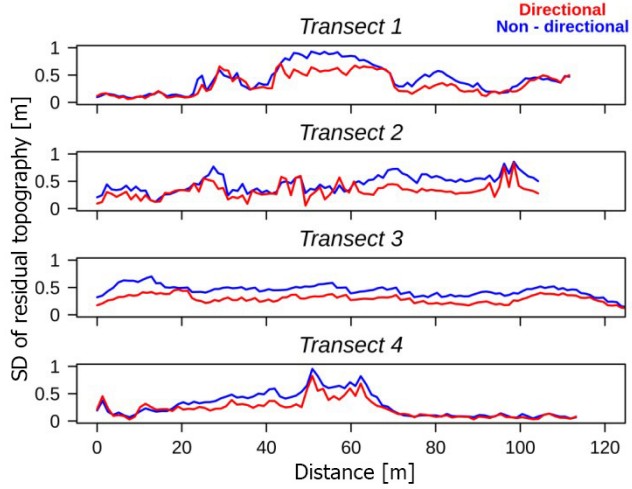


**Figure 7: Analysis of the four transects from the Braema study area using the *SD of residual topography algorithm* (red) or without (blue) direction in the calculation.**

### 3.3 Case study: snow avalanche modelling

Calculated surface roughness differed strongly when a DSM was used as input data instead of a DTM (Fig. A4 in Appendix).
In dense forest and in a windthrow area the calculated surface roughness was overestimated and it depicted mostly the tree crowns or branches of the lying logs. Surface roughness calculated from the DSM considered the uppermost surface features, in comparison to calculations based on the DTM, where only terrain was considered and all the surface features were filtered out. The DTM-derived roughness values were thus lower overall compared with the DSM-derived values, in particular in the presence of forest vegetation and in the windthrow areas (Fig. A4 in Appendix).

The roughness difference between DSM and DTM has important implications for the numerical simulation of gravitational mass movements, as illustrated in the avalanche simulation based on LiDAR data. Simulations performed using the DSM resulted in a 25% (Braema) and 14% (Franza) shorter runout distance and a more dispersed flow pattern than those based on the DTM (Fig. 8 and Fig. A5 in Appendix). When using the DSM, we identified the interaction between the snow mass and the features on and above the ground, such as sparse forest and the windthrown areas. The maximum flow height based on the
DSM was therefore 0.4 m and 0.2 m greater for Braema and Franza, respectively, compared with values based on the DTM. Using the DSM, the runout distance decreased by 112 m in Braema and by 20 m in Franza. As shown in Fig. 8, the snow mass did not impact the forested areas and there was no tree destruction in the simulation. However, there was a visible interaction between the avalanche and the sparse trees in the runout area in the simulation based on the DSM.

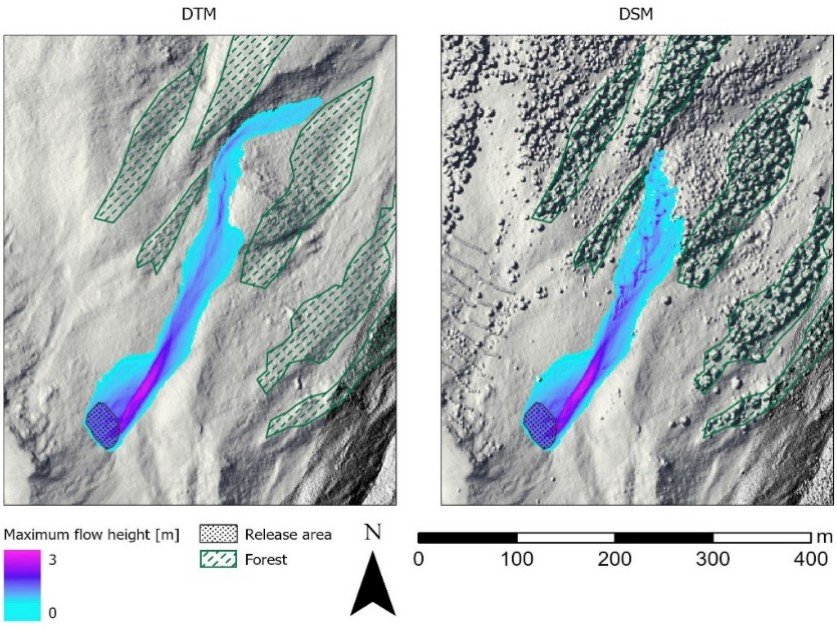

**Figure 8: Avalanche simulation output (maximum flow height) in the Braema study area. The avalanche runout distance was 112 m**
**greater when the DTM was used as the input model for the simulation than when the DSM was used (visualized are the hillshades calculated from the terrain and surface model swissALTI3D; swisstopo, 2018). The maximum flow height was 0.4 m greater when the DSM was used, as a result of the interaction with the roughness features.**



## 4 Discussion

### 4.1 Roughness classification and algorithm evaluation

We tested seven algorithms for calculating surface roughness, with three spatial resolutions and three moving window areas, for terrain classification. The best performing algorithm was the *vector ruggedness measure*, which distinguished between the roughness categories in an accurate way for all of our tested resolutions (0.1 m, 0.5 m and 1 m) and moving window areas (9 m$^{-2}$, 25 m$^{-2}$ and 49 m$^{-2}$). However, the performance did not increase with higher resolution. This is probably due to the scale of our features of interest. These features are not that detailed as the higher resolutions of 0.1–0.5 m might be able to

distinguish. *SD of profile curvature*, *SD of residual topography* and *SD of slope* were also accurate in distinguishing between the roughness categories. The fewest errors across all algorithms occurred with the resolution of 1 m, where only *area ratio*, *SD of slope* and *vector dispersion* did not correctly classify some of the roughness categories (one error for *area ratio* and *SD slope* and two errors for *vector dispersion*). The lowest spatial resolution (1 m) delivered the best results, offering a reliable basis for roughness classification on larger scales. DEMs with higher resolutions, such as 0.1 m and 0.5 m, are not as

widespread as the 1 m resolution DEMs that are commonly available for large areas of the Alpine region. Moreover, interpretations of analysis based on data from larger areas will not be affected by potential errors in DEMs (Riley, 1999). The best performing combination of spatial resolution and moving window area was 1 m and 49 m$^{-2}$ (with no pairs of overlapping distributions), which was the lowest resolution and the largest moving window area in our analysis. The use of higher resolution models (< 1 m) had no additional advantage in our study, which is in line with findings from other studies (López-Vicente and

Álvarez, 2018; Yang et al., 2014). Moreover, this result is relevant to large-scale risk evaluation and analysis, since digital models with a resolution < 1 m are not so frequent.

All the tested algorithms had at least one combination of spatial resolution and moving window area without a pair of overlapping distributions. The most suitable algorithm for an investigation thus depends on its purpose and the land-cover types involved. In two of the tested combinations of resolution and moving window area (0.1 m and 49 m$^{-2}$, 0.5 m and 25 m$^{-2}$),

the algorithm *SD of curvature* failed to distinguish between windthrow areas and high forest. With some of the combinations, three of the algorithms (*area ratio*, *SD of residual topography* and *vector dispersion*) did not distinguish between shrub forest and windthrow, due to the similar height and structure of these categories. If an extensive evaluation of different resolutions and moving windows is not realistic in an investigation, we suggest using the roughness algorithm *vector ruggedness measure* because it showed the best performance overall in distinguishing between the roughness categories, in

particular with the 1 m resolution DSM and a moving window area of 49 m$^{-2}$.

The most common difficulties were in distinguishing between rocky and smooth or very smooth terrain. The algorithm *area ratio* failed in all combinations for the spatial resolutions of 0.1 and 0.5 m, with overlapping distributions for rocky and smooth terrains and for rocky and very smooth terrain in all three combinations of 0.5 m resolution. The algorithm *terrain ruggedness index* also failed in distinguishing between rocky and very smooth or smooth terrain in all combinations of 0.5 m resolution.

Both of these algorithms assigned higher roughness values to the categories very smooth and smooth compared with the other





algorithms. Grohmann et al. (2011) also found that *area ratio* showed higher values for the smooth slope of a scarp, highlighting a major disadvantage of this algorithm in that smooth steep slopes can be classified as rough. This might be an important point for choosing the right algorithm in natural hazard mapping.

We calculated thresholds for distinguishing between the roughness categories, which may be further used in roughness classifications of other areas. These categories are a novelty in the literature and they can be considered a preliminary proposal. In fact, they must be applied carefully, as the surface of snow can be highly variable (Buhler et al., 2016). In our study, the snow surface consisted of remaining snow patches in summer and was very smooth, as shown with the lowest distribution of roughness values (Fig. 4).

### 4.2 Surface roughness in natural hazard modelling

The assessment of surface roughness can lead to better estimation of potential avalanche release areas (Bühler et al., 2018), as well as improving avalanche simulations by including areas with high roughness values (as RAMMS additional friction areas). In Bühler et al. (2018) the model input for large-scale avalanche release area delineation is a DTM, from which vegetation and other features are removed. Forests are handled as a binary layer so that potential release areas covered with dense forests are excluded (Bühler et al., 2018). This approach may, however, underestimate the protection function of sparse and young forests

that are not officially classified as forest but influence the natural hazard dynamics for modelling purposes. In the case of avalanches, lying logs after a windthrow event may support the snow and contribute to the stabilization of the whole snowpack, similar to the function of shrub forests or young forests (Bebi et al., 2009; McClung and Schaerer, 2006; Schneebeli and Bebi, 2004; Teich et al., 2012a; Wohlgemuth et al., 2017). However, in the case of avalanches or debris flows releasing above the forest, fallen trees may be entrained rather than slowing or stopping the avalanche/debris flow, as in the case for young forests

(Ishikawa et al., 2000; Michelini et al., 2017; Teich et al., 2013). In debris flow simulations, surface roughness is an important input parameter in extreme scenarios, in which the flow may spread outside the main channel, flooding different terrains such as roads, rocky areas, and young and old forests (May, 2002). In this case appropriate models have to be selected with friction parameters that can be spatially distributed to include the influence of roughness (Hungr and McDougall, 2009; Mergili et al., 2017). Addittionally, roughness classification can quantify surfaces affected by a land-use change (e.g. windthrown forest or

shallow landslides), identifying new potential sediment source areas such as shallow landslides (Huebl and Fiebiger, 2007). Terrain roughness and its classification can increase both the accuracy of natural hazard simulations and the preliminary identification of potentially dangerous areas that require accurate evaluation.

### 4.3 Directional roughness

In order to further improve the applicability of roughness categories we implemented a directional surface roughness approach.

This approach helped us to better represent the surface roughness along the mass flow direction, with results that were substantially different from values assigned from a topographical point of view. Normally, gullies are considered rough using a non-directional algorithm, but they can be smoother in the direction of the dominant natural hazard flow. The directional





surface roughness approach, which was available for all the tested algorithms using standard deviation, yielded lower values for roughness along the flow direction. In our study area Braema, this resulted in a more realistic assignment of channelized

gullies roughness, which would be categorized as very rough in a standard roughness map. Implementing directional roughness thus seems to result in more realistic results. A further improvement for surface roughness within gullies would be an automatic identification of gullies and an application of the directional algorithm automatically for a buffer area along the gullies, therefore improving the roughness maps.

## 4.4 Applications

In our study we addressed relevant land-cover types in mountain forests and treeline ecotones of the southern and central Alps. While we expect that our results are also relevant for similar ecosystems characterized by coniferous forests, comparable analysis and a verification of the classification would be necessary in order to further generalize our results. Similarly, this would be required for the classification of other disturbed forest stands (e.g. after a bark beetle outbreaks or wild fires), since different disturbances with different intensities create particular structures that most likely have unique patterns of surface

roughness (Franklin et al., 2002; Hansen et al., 2016; Waldron et al., 2013).

We selected the Franza study area in order to analyse a disturbed forest immediately after a windthrow. The forest protection function is altered when a forest is disturbed. Therefore, there is a need for practitioners to assess the protection capacity of the remaining structures on the ground for natural hazard mapping. In the case of snow avalanches, the very small number of avalanches observed after these disturbances indicates that lying logs contribute to increased terrain roughness and thus to the

conservation of a considerable protective function against avalanches, at least for the first two decades after a disturbance event such as windthrow (Wohlgemuth et al., 2017). In the same way, early successional stages of post-disturbance development can provide effective protection in avalanche release zones. However, these structures are usually not classified as forest stands, since in most of cases they do not match the minimum criteria defined by the authorities (i.e. density, mean height; Brändli and Speich, 2007; FAO, 2015; INFC, 2005), so these structures might not be included in the definition of potential avalanche

release areas. Lying deadwood can also provide a residual protective function for rockfall. Thanks to the higher impact probability compared with standing trees, the flexibility of the logs on the ground, disturbed forest areas can reduce the rock velocity and absorb kinetic energy (Bourrier et al., 2012; Ringenbach et al., 2021). This is especially the case in the first phase after a disturbance, when the decaying processes have not yet reduced the wood strength (Amman, 2006).

The analysis of surface roughness could therefore serve as a useful proxy for assessing the hazard risk in disturbed forests, but

it has some limitations as well. By analysing surface roughness over time, one could additionally observe landscape transformations and changes in vegetation (natural or anthropogenic) that affect surface roughness and consequently natural hazards processes. In particular, by calculating surface roughness for different vegetation types, snow gliding could be easily modelled and predicted for different land-use scenarios. This could improve the identification of areas exposed to natural hazards and aid in the implementation of protection measures (Leitinger et al., 2008). In the case of old disturbed forest, a

roughness time-series analysis might not distinguish between the roughness of old lying logs, lower vegetation and tree



regeneration. After years of decomposition, the lying logs become less supportive, decrease in height, are moved and even decompose completely (Bebi et al., 2015; Wohlgemuth et al., 2017). A comprehensive overview of the decay process over a longer period after a disturbance (more than 20 years) would be helpful to understand the function of time and the remaining protection capacity after a disturbance such as windthrow. However, considerable variability across different environmental
gradients may occur, and every area should therefore be handled individually, especially if elements of risk exist. Thus, a combination of calculated surface roughness and field investigations may be necessary in such areas (e.g. windthrown forest or large landslides), where an accurate evaluation of the ground features cannot be performed by a DEM survey alone.

For the estimation of avalanche release areas and avalanche propagation, even small-scale topographic roughness can have an influence on the runout distance of ground-releasing processes, as in the case of wet snow avalanches (Sovilla et al., 2012).
This is also important for small avalanches with small release depths and a shallower snowpack (McClung, 2001), since very large snow depths can bury the surface roughness and therefore smoothen the surface (Veitinger et al., 2014). Using DSMs could improve the surface roughness estimation, as demonstrated with the *vector ruggedness measure* algorithm in our study. It had no pairs of overlapping distributions for all the roughness categories, and it accurately assigned high roughness values to higher vegetation, avalanche barriers and other land-cover categories. In comparison, the DTM-based approach generally
underestimated the surface roughness (Brožová et al., 2020). The case study, applying numerical avalanche modelling to a DSM and a DTM, showed that surface roughness plays a decisive role in the avalanche runout distance and the flow path. However, in the case of high and dense forests, the surface roughness classification based on DSM is limited. The surface roughness values calculated from the DSM represent the tree crowns, which are classified as rough. But the crowns usually do not interact with an avalanche flow (except powder snow avalanches). Therefore, DTMs should be applied to calculate the
surface roughness within dense forests and DSMs should only be applied for open areas, where roughness may still interact with the hazard process but is not included in the forest classification. In this way, areas with increased roughness outside of defined forest areas could be detected and included within the hazard modelling. In the case of avalanches, the RAMMS simulation tool (Christen et al., 2010) offers a possibility to add an area with increased friction parameters. A smart combination of DSM and DTM data may result in better estimation of the surface roughness faced by the gravitational mass
movement.

## 5 Conclusions

Our study shows that DEMs with a spatial resolution of 1 m, which are becoming increasing available, are well suited for use with roughness algorithms for natural hazard terrain classification and that higher spatial resolutions (0.1–0.5 m) do not necessarily improve the terrain surface roughness classification.
From our tested algorithms, *vector ruggedness measure* showed the best performance in distinguishing between different roughness categories. However, depending on the study area and relevant land-cover types, it is also possible to use other algorithms, with careful choice of spatial resolution and moving window area. In order to avoid overestimation of terrain





roughness for natural hazard applications in study areas where mass flow is continuously confined, we suggest applying the directional roughness approach. This improvement is available for any of the algorithms using a standard deviation, e.g. *SD of*

*residual topography*.

Considering terrain roughness with an appropriate algorithm and in a specific spatial context may improve the generation of forest layers applied for large-scale hazard indication mapping. In particular, smaller protection forest stands, which are currently underrated and poorly investigated, could be better represented.

Finally, using DTMs in combination with DSMs may further improve the modelling of natural hazards. In fact, based on well

descriptive surface roughness maps, practitioners could identify and successively analyse areas where the implementation of protection measures is necessary to mitigate potential hazard consequences for people and infrastructure.



# 7 Appendix

Roughness categories: 1 - snow, 2 - very smooth, 3 - smooth,
4 - shrub forest, 5 - high forest, 6 - rocky, 7 - windthrow


**Figure A1: Distribution of roughness values according to different roughness categories (1 – snow, 2 – very smooth, 3 – smooth, 4 – shrub forest, 5 – high forest, 6 – rocky, 7 – windthrow) for seven algorithms (*area ratio*, *vector ruggedness measure*, *SD of profile curvature*, *SD of residual topography*, *SD of slope*, *terrain ruggedness index* and *vector dispersion*) for the spatial resolution of 0.1 m. Red arrows show the overlapping distribution for a pair of categories that the given algorithm fails to distinguish.**




**Figure A2: Distribution of roughness values according to different roughness categories (1 – snow, 2 – very smooth, 3 – smooth, 4 – shrub forest, 5 – high forest, 6 – rocky, 7 – windthrow) for seven algorithms (*area ratio*, *vector ruggedness measure*, *SD of profile curvature*, *SD of residual topography*, *SD of slope*, *terrain ruggedness index* and *vector dispersion*) for the spatial resolution of 0.5 m. Red arrows show the overlapping distribution for a pair of categories that the given algorithm fails to distinguish.**





**Figure A3: Calculated surface roughness in the study area Franza using the seven investigated algorithms:** *area ratio* **(1),** *vector ruggedness measure* **(2),** *SD of profile curvature* **(3),** *SD of residual topography* **(4),** *SD of slope* **(5),** *terrain ruggedness index* **(6) and** *vector dispersion* **(7). The same area is presented as an orthophoto (8) (drone flight, 2019) and in DSM hillshade (9). All algorithms**
**were calculated based on the overall best performing combination of spatial resolution (1 m) and neighbourhood (moving window 7 × 7 m). To improve the visualization and compare the roughness maps, we normalized them with the 25th percentile as the minimum value and the 75th percentile as the maximum one.**





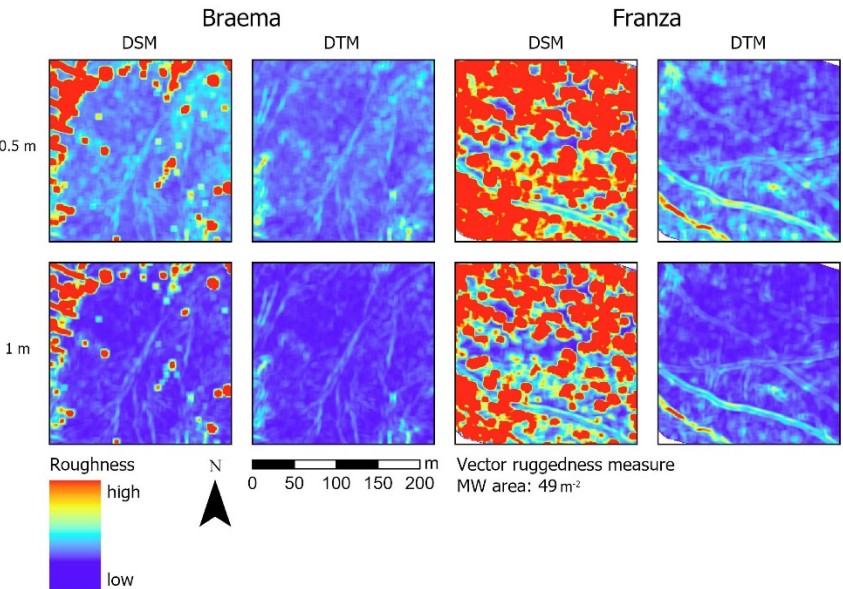

**Figure A4: Calculated surface roughness in the two study areas Braema and Franza, using DSM and DTM and the *vector ruggedness measure* algorithm.**

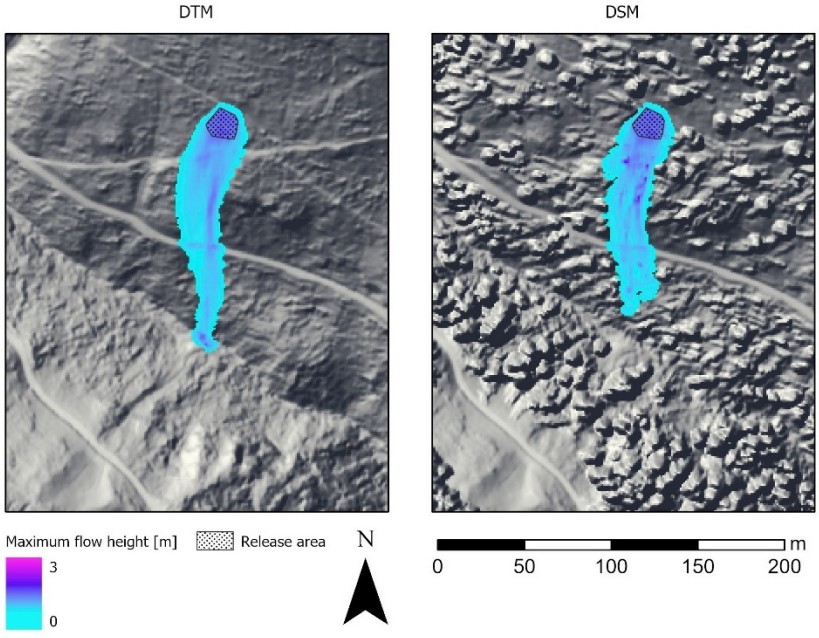

**Figure A5: Avalanche simulation output (maximum flow height) in the Franza study area. The avalanche runout distance was 20 m longer when the DTM was used as the input model for the simulation than when the DSM was used (LiDAR data provided by the region Veneto). The maximum flow height was 0.2 m greater when the DSM was used, as a result of the interaction with the roughness features.**



## 8 Code and data availability

The code for computing the terrain roughness is available at the following link:

https://github.com/TommBagg/terrain_roughness_GRASS.git

Data are available from the corresponding author on request.

## 9 Author contribution

NB conceptualized the research, performed the statistical analysis and RAMMS simulations, and wrote the manuscript; TB conceptualized the research, implemented the terrain roughness algorithms in code and wrote the manuscript; VdA reviewed the manuscript and suggested the directional algorithm; YB reviewed the manuscript and structured the LiDAR roughness computation and RAMMS simulations; PB defined the research structure and reviewed the manuscript.

## 10 Competing interests

The authors declare that they have no conflict of interest.

## 11 Acknowledgements

This research received funding as a part of the project "Bridging the mass-flow modelling with the reality", from the CARIPARO foundation (2724/2018), from the WSL research programmes CCAMM and SwissForestLab and from the
prevention foundation of the Swiss cantonal building insurance (KGV). The authors wish to thank Lorenzo Martini (TESAF Department) for providing the photogrammetric survey (drone flight) of the Franza study area and to Melissa Dawes for finalizing our manuscript and improving the language clarity and quality.





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
