# Peer review of "Multiscale analysis of surface roughness for the improvement of natural hazard modelling"

_Natural Hazards and Earth System Sciences, 2021_

## Author Response (AR1)

**Response to the Reviewer #1**

Dear Manuel López-Vicente,

Thank you very much for your comments and suggestions to our paper. We answer your comments below.

- *Abstract. Include the method/s used to generate the DSM and the DEM, e.g. LiDAR, SfM. The type of method influences the point density, and thus, the ability of the obtained models to accurately capture macro-, meso- and micro-features of the landscape.*

We will add the following information to the abstract (new information **in bold**).

"... we tested seven roughness algorithms using **photogrammetric** digital surface models (DSM) with different resolutions …"

"We simulated avalanches on different elevation models (**LiDAR-based**) to observe a potential influence of a DSM and a digital terrain model (DTM)."

- *Abstract. Authors compare the results of "surface roughness" based on DSM vs. those obtained from DTM. I have serious doubts that this is correct. In a DSM, all landscape features are included, and thus, the corresponding values of surface roughness are associated to those features. However, the information captured in a DTM is mainly controlled by the ground elevation, without most of the features included in the DSM. From a strict point of view, a DTM is the sum of the DEM and the main landscape geomorphic features like rivers, cliffs, crests, and breaking points. Therefore, the surface roughness derived from a DSM and a DTM of the same site is always qualitatively different. Besides, the digital height model (DHM=DSM-DTM) obtained in different land uses is different because of the distinct features that characterise each land use. All these aspects should be clarified in a revised version.*

Thank you for this valuable point. We showcase the differences between the DSM and DTM in order to emphasize the importance of surface roughness in the avalanche simulation. Above treeline, the DSM is very similar to the DTM except for small bushes, young trees and man-made structures. As soon as we get into forested terrain, the differences are becoming greater. Forests are already implemented within the RAMMS simulations. However, the sparse vegetation or other surface roughness features above the treeline are not represented within the simulations, since the DTM is used. As surface features like sparse vegetation are removed from the DTM, we believe that there may be an overestimation of the simulated runout. When we simulate the snow avalanche on the DSM, it interacts with the surface features included in the model and therefore better represents the reality. In fact, hazard processes show different behaviours when they interact with surface features. Based on our study, we do thus not propose to simulate hazard processes using a DSM instead of the DTM, but we stress the importance of surface roughness on different simulations. For this reason, we propose to accurately select the adequate surface representation (DSM, DTM or a combination of them) for the calculation of surface roughness, which should be included in modelling of hazard processes.

- *Abstract. What is the novel aspect of this study? I suggest highlighting the actual contribution of this study based on the available literature. To clearly present these aspects will make the article more attractive for potential readers.*

As stated above, the importance of surface roughness for natural hazard modelling is great (shown on the example of a snow avalanche). We found that some widely used surface roughness algorithms based on the DSM could be used to assess the roughness in alpine terrain. Together with the directional roughness this is a promising approach for achieving better assessments of surface roughness, which should be included in the hazard modelling in the future.

In order to make this point clearer in the abstract, we rephrased the end of the abstract:

"We simulated avalanches on different elevation models **(LiDAR-based)** to observe a potential influence of a DSM and a digital terrain model (DTM) **using simulation tool RAMMS. In this way, we accounted for the surface roughness based on a DSM instead of a DTM, which resulted in shorter simulated avalanche runouts by 16–27% in the two study areas. Surface roughness above a treeline, which in comparison to the forest is not represented within the RAMMS, is therefore underestimated. We conclude that using DSM- in combination with DTM-based surface roughness and considering the directional roughness is promising for achieving better assessment of terrain in alpine landscape, which might improve the natural hazard modelling.**"

- *L.33-34. Please, provide more information of the six cited studies or choose the three most relevant studies. In my opinion, it is not necessary to include six articles to support one statement. Do the same in L.42-43 with the 5 references: include more information of the modelling approaches*

We kept the following cited studies as they represent the wide field of possible applications of surface roughness.

"Quantifying surface roughness is thus central for the estimation of various biophysical characteristics and ecosystem services (; Koponen et al., 2004; ; Wu et al., 2018). "

We kept the following studies to support the statement since they represents two advanced model to simulate flow propagation. Furthermore, we included more information involving modelling approaches.

"Approaches to modelling flow propagation are numerous (; Frank et al., 2017; ; Pudasaini and Mergili, 2019). **They can represent the flow as a single-phase or a multi-phase consisting of solid and water component propagating through a given topography (Christen et al., 2010; Rosatti and Begnudelli, 2013)**. **Some of them include a spatial variability of the friction parameters and can even simulate erosion processes (Hungr and McDougall, 2009; Mergili et al., 2017).**"

- *L.102-104. These two sentences are very interesting. I suggest authors extend the explanation of this approach.*

We include the following paragraph in the revised manuscript:

"However, the investigated natural hazards have a predominant diffusion direction identified as the combination of terrain slope and curvature. Some studies implemented the surface roughness along a predefined direction (Michelini, 2016; Trevisani and Rocca, 2015). **The direction for which roughness has been computed, usually derived through GIS algorithm (D8 or D-infinity), applied to the original or smoothed digital models. However, the direction derived through neighbourhood cells analysis could not be the same of the mass flow propagation. Such behaviours may be observed when the routing volumes are extreme and therefore in some particular situations the propagation direction may be defined by its inertia rather than the topography (Guo et al., 2020). In other cases, the particular mountain topography may force mass flows to affect the opposite hillside of the valley through a runup mechanism (Iverson et al., 2016). Furthermore, the flow direction of banks and channel sides features computed with GIS algorithms do not usually correspond to the mass flow direction. In this situation bank direction can be improved through a smoothing process of the DTM in order to remove gullies and channel from the basal topography. This technique can be easily applicable in case of regular channels but it could become more complex when the channel morphology is irregular, since it could oversimplify the basal topography. For such reasons in this study, we propose a novel approach to calculate surface roughness along user defined lines.**"

- *L.134: Which features were included in the DTM? I assume that all landscape features were represented in the DSM. However, it is not clear which features were included in the DTM, apart from the DEM. This is an important aspect, because the derived products from the DSM and DTM -as the surface roughness and terrain roughness- are qualitatively different, and thus, it has no sense to compare them in a direct way.*

As answered already in the second comment (related to the comment on the abstract), we compare the DTM and DSM in order to stress the importance of surface roughness and different manifestations of surface roughness in DTM and DSM for natural hazard modelling. The DSM represents all landscape features, while in the DTM the vegetation cover (represented by trees, deadwood and shrubs) and man-made structures are removed. By using these two surface models, we show how is the avalanche flow influenced by the surface features. The DSM derived roughness adequately captures such features which are filtered out and missing in the DTM.

- *Section 2.1 and Figure 1. What is the criteria followed and the method used to draw the boundaries of the two study areas?*

The boundaries depend on the availability on high quality drone data in the two study areas.

- *L.156. Add the country of the SenseFly company, as you did it in line 171 with DJI.*

The area of almost 1 km$^{-2}$ was surveyed on 17 June 2019 using a senseFly eBee+ drone (**Lausanne, Switzerland**) equipped with an RTK GNSS system for accurate georeferencing (better than 5 cm).

- *Discussion. I would like to know the average size of the landscape features included in the analysis of this study. Besides, authors should evaluate the relationship between the size of the features and the extension of each window area. Maybe, if some features are much smaller than the window area, the information of those features is blurred. Maybe, the suitable window area may depends on the average dimensions of the features. This idea should be discussed.*

We add the following sentences in the revised manuscript:

"However, the performance did not increase with higher resolution. This is probably due to the scale of our features of interest. **Features in our study areas like shrubs, rocks, standing or laying trees, but also gullies are usually in the scale of meters.** These features are not that detailed as the higher resolutions of 0.1–0.5 m would be able to distinguish."

"**In our study, we could not find a relationship between the size of the roughness features (in meter scale) and the size of the moving window area. The best performing moving window area was analysed as the largest tested – 49 m$^{-2}$, in combination with the 1 m resolution.**"

- *Discussion. Did you establish the thresholds for distinguishing between the roughness categories before running the algorithms or after obtaining the results? This aspect has to be clearly explained, and the numerical criteria to propose those thresholds too.*

Thank you for your observation, we have this information in the methods part and will point it out in the discussion as well.

We rephrased this in the methods part 2.3 Design and statistical analysis of roughness categories, line 250:

"In order to obtain a classification based on threshold values for a technical purpose, we analysed the kernel density distribution between the roughness categories (Table 2)**, after evaluating the best-performing algorithm,** to determine the point of minimum overlap. We used the overlap function (overlapping package; Pastore, 2018; Pastore and Calcagnì, 2019) implemented in R (R Core Team, 2021). This intersection is the threshold between two roughness categories (*xpoints*)."

We also rephrased this in the discussion part, line 424:

"**After finding the best-performing algorithm (*vector ruggedness measure*),** we calculated thresholds for distinguishing between the roughness categories, which may be further used in roughness classifications of other areas. These categories are a novelty in the literature and they can be considered a preliminary proposal. **However, these values must be applied carefully, as they were assigned using the vector ruggedness algorithm based on the 1 m-resolution DSM and moving window area of 49 m$^{-2}$. One should be as well cautious when defining the roughness categories as e.g. the surface of snow can be highly variable (Bühler et al., 2016).** In our study, the snow surface consisted of remaining snow patches in summer and was very smooth, as shown with the lowest distribution of roughness values (Fig. 4). **We therefore propose further validation of such values over larger areas and different landscapes.**"

**Response to the Reviewer #2**

Dear Referee 2,

Thank you very much for your comments and suggestions to our paper. We answer your comments and questions below. New parts added to the manuscript are in **bold** and erased sentences in .

1- *Now that the author has realized that the spacing of DSM will affect the results of terrain analysis, especially the primary terrain attributes: local slope gradient, roughness, curvature. In fact, the relevant study has existed in this field for decades. Therefore, why didn't the author use the Root mean square slope (Hutchinson, 1996) to find the optimal resolution at the beginning? In this way, a lot of calculation costs can be saved, and different land uses and topography should be suitable for different resolutions. Plus, a fine spatial resolution of DSMs is no longer an issue, as the author mentioned in the introduction.*

One of the aims of our study is to test the use of high-resolution DSMs for the calculation of terrain roughness indices. For this purpose, we selected three spatial resolutions. For our study we did not use the Root Mean Square Slope reported in Hutchinson (1996), since the surface roughness indices that we analysed are directly influenced by the DEM resolution. In fact, lower DEM resolution (equal to higher accuracy) resulted in a better representation of surface features (Glenn et al., 2006). Different studies indicated that an accurate evaluation of DEM resolution is necessary to correctly identify the features or processes of interests (Deng et al., 2007; Grohmann et al., 2011; Habtezion et al., 2016; Keijsers et al., 2011; Tarolli and Tarboton, 2006). Since the features of interest of our study have different sizes (trees, rocks, deadwood, disturbed forests, shrubs) we tested seven roughness algorithms looking for the best combination of DEM resolution and moving window size to be used in natural hazard modelling to better differentiate terrain classes that affect simulation of hazard processes.

2- *I have a high interest in directional roughness, and I would be happy to see related calculation methods and literature reviews.*

In the reviewed version of the manuscript we incorporate the following paragraph (new parts are reported in bold).

Line 102 "number of neighbourhood cells. **In such sense, most of the roughness indices reported in literature considered the DEM as an isotropic surface. However, the concept of surface anisotropy is of fundamental importance for the investigation of geomorphological features and channelized or dispersed flows (Busse and Jelly, 2020; Insua-Arévalo et al., 2021; Middleton et al., 2020). If the surface shows an anisotropic texture the flow resistance is directly influenced by obstacles disposed along the flow direction. Since,** the investigated natural hazards show a predominant

diffusion direction identified as the combination of terrain slope and curvature, **texture anisotropy has to be taken in account when simulating mass flows (Roy et al., 2016; Viero and Valipour, 2017)."**

Here we would continue with the reply of the comment raised by reviewer 1. We report the paragraph below.

" **Some studies implemented the surface roughness along a predefined direction** (Michelini, 2016; **Trevisani and Rocca, 2015**). **The direction for which roughness has been computed, usually derived through GIS algorithm (D8 or D-infinity), applied to the original or smoothed digital models. However, the direction derived through neighbourhood cells analysis could not be the same of the mass flow propagation. Such behaviours may be observed when the routing volumes are extreme and therefore in some particular situations the propagation direction may be defined by its inertia rather than the topography (Guo et al., 2020). In other cases, the particular mountain topography may force mass flows to affect the opposite hillside of the valley through a runup mechanism (Iverson et al., 2016). Furthermore, the flow direction of banks and channel sides features computed with GIS algorithms do not usually correspond to the mass flow direction. In this situation bank direction can be improved through a smoothing process of the DTM in order to remove gullies and channel from the basal topography. This technique can be easily applicable in case of regular channels but it could become more complex when the channel morphology is irregular, since it could oversimplify the basal topography. For such reasons in this study, we propose a novel approach to calculate surface roughness along user defined lines**."

> 3- *From figure 5, there seem to be two clusters of results, one group consisting of area ratio, SD of residual topography, terrain ruggedness index, and vector dispersion. This group has lost a lot of details, especially in the lower-left corner of the image. I would like to see the explanations of the results (difference) of these seven algorithms firstly.*

In the lower left corner we see avalanche barriers, which are man-made structures for protecting against avalanche release. They are usually up to 5 m high and from above (from the orthophoto) with less than 1 m width. We believe, that this might be the reason for wrong interpretation by some of the algorithms. We present these results in the section 3.1 Roughness classification and algorithm evaluation and we add these lines to the first paragraph of the 4 Discussion (new phrases in bold):

3.1 Roughness classification and algorithm evaluation

"Surface roughness calculated with the seven different algorithms and normalized using the same colour range (Fig. 5 and Fig. A1 in Appendix, for the Braema and Franza study areas) revealed important differences in the ability to identify specific terrain and vegetation types. As visible for the overall best performing combination of resolution and moving window (1 m and 49 m$^{-2}$) in Fig. 5, all algorithms distinguished accurately between high vegetation (forest) and other vegetation types. Nevertheless, some of the algorithms (*vector dispersion, SD of residual topography, **terrain ruggedness index** and area ratio*) failed to detect the avalanche barriers correctly and falsely identified them as rather smooth. Also, small gullies were not clearly separated with some of the algorithms and were particularly poorly visible with the algorithms *SD of profile curvature* and *SD of slope*, whereas they were successfully identified with moderate roughness values by the other algorithms. Smooth surfaces were visualized with lower roughness values (darker blue in Fig. 5) by algorithms like *vector ruggedness measure, SD of residual topography* and *vector dispersion* [Fig. 5 (2, 4 and 7)]. Other algorithms [Fig. 5 (1, 3, 5 and 6)] assigned these smooth surfaces rather high roughness values (lighter blue to cyan blue in Fig. 5)."

4 Discussion

Grohmann et al. (2011) also found that area ratio showed higher values for the smooth slope of a scarp, highlighting a major disadvantage of this algorithm in that smooth steep slopes can be classified as

rough. **The algorithms vector dispersion, SD of residual topography, terrain ruggedness index and area ratio could not detect the avalanche barriers in the study site Braema. This might be due to small width (less than 1 m) of these objects together in combination with the relatively large moving window area (49m⁻²). Such issues might play an important role**  for choosing the right algorithm in natural hazard mapping.

4- *In the 4.3 Application, I read it several times. It is really difficult to follow the author's logic. I still don't know how the author wants to apply the results. I can vaguely know that the author wants to apply to the ecosystem, but how?*

The study highlights the importance of surface roughness in the simulation of natural hazards. The study evaluated the best performing algorithms for land cover classification. The identified classes represent ground features influencing the mass flows. Applications of the study can be straight applied to improve the reliability of model outcomes. Furthermore, we emphasize the fact to adequately represent land cover characterized by disturbed forests as they usually are not correctly implemented.

We changed the title of the section 4.4 Applications **for natural hazard assessment** to state the focus of the possible application. We further modified this section to make it more comprehensible. Below we report the section and in bold the part we add to the revised version.

"In our study we  **classified** relevant land-cover types  **of** mountain forests and treeline ecotones of the southern and central Alps. **The classes represent land cover characterized by features that influence mass flows propagation in different ways. The derived roughness maps or classes could be straight used in order to improve the reliability of simulation models. Since we analysed two alpine areas,**  **we can assume** that our results are also relevant for similar ecosystems characterized by coniferous forests **However,** comparable analysis and a verification of the classification would be necessary in order to further generalize our results. Similarly, this would also be required for the classification of other disturbed forest stands (e.g. after a bark beetle outbreaks or wild fires), since different disturbances with different intensities create particular structures with expected diverse surface roughness (Franklin et al., 2002; Hansen et al., 2016; Waldron et al., 2013).

**Moreover, the surface roughness classification and the selected roughness algorithm included the identification and analysis of a forest damaged by a wind storm: Franza case study.**  In cases comparable to this, the forest protection function is altered, when a forest is disturbed. Therefore, there is a need for practitioners to assess the protection capacity of the remaining structures on the ground for natural hazard mapping. In the case of snow avalanches, analysis of field data as well as the very low number of avalanches observed after these disturbances indicate that lying logs contribute to increased terrain roughness and thus to a conservation of a considerable protective function against avalanches at least for the first two decades after the disturbance event as windthrow (Wohlgemuth et al., 2017). In the same way, also early successional stages of post-disturbance development can provide a good protection in avalanche release zones. However, these structures are usually not classified as forest stands, since in most of the cases, they do not match the minimum criteria defined by the authorities (i.e. density, mean height; (Brändli and Speich, 2007; FAO, 2015; INFC, 2005) so these structures might not be included for the definition of avalanche potential release areas. The lying deadwood can also still provide a residual protective function for rockfall. Thanks to the higher impact probability compared to standing trees, the flexibility of the logs on the ground, disturbed forest areas can reduce the rock velocity and absorb kinetic energy (Bourrier et al., 2012; Ringenbach et al., 2021). Especially as in the first phase, when the decaying processes have not reduced the wood strength (Amman, 2006). **Therefore, in this study we included in the surface roughness analysis and classification these land cover types (disturbed forests, young forests and shrubs) that are usually not adequately evaluated for natural hazard modelling.**

The analysis of surface roughness could therefore serve as a  **good** proxy **to evaluate some of the hazard temporal evolution**  in disturbed forests, but it has some limitations as well. By analysing surface roughness over time, we could also observe landscape transformations and change in vegetation (natural or anthropological) that affect surface roughness and consequently natural hazards processes. In particular, calculating surface roughness for different vegetation types, snow gliding could be easily modelled and predicted for different land-use scenarios. This could improve the identification of natural hazards exposed areas and further implementation of protective measures (Leitinger et al., 2008). In the case of old disturbed forest, the roughness time series analysis might not distinguish between roughness of the old lying logs, lower vegetation and tree regeneration. After years of decomposition, the lying logs become less supportive, decrease in height and they even displace (Bebi et al., 2015; Wohlgemuth et al., 2017). A comprehensive overview of the decay process in longer period after a disturbance (more than 20 years) would be helpful to understand the function of time and the remaining protection capacity after a disturbance such as windthrow. However, a great variability across different environmental gradients may occur, therefore every example should be handled individually, especially if elements of risk exist. Thus, a combination of calculated surface roughness with field investigations may be necessary in such areas (e.g. windthrown forest or large landslides), where an accurate evaluation of the ground features cannot be performed by a DEM survey only.

**Surface roughness further influences**  the estimation of avalanche release areas and avalanche propagation. Even a small-scale topographic roughness may have an influence on the runout distance of ground-releasing processes as the case of wet snow avalanches (Sovilla et al., 2012). This is also important for small avalanches with little release depths and shallower snowpack (McClung, 2001), since very high snow depths may burry the surface roughness and therefore smoothen the surface (Veitinger et al., 2014). Using DSMs **for terrain representation in models** could improve the surface roughness estimation as showed on the example of the Vector ruggedness measure in our study. It had no pairs of overlapping distribution for all the roughness categories and it assigned well the rough values for higher vegetation, avalanche barriers and other land cover categories; compared to the roughness calculated from a DTM, which have generally underestimated the surface roughness (Brožová et al., 2020). The case study, applying numerical avalanche modelling on DSM and DTM, showed that surface roughness plays a decisive role for the avalanche runout distance and the flow path.  In the case of high and dense forests, the surface roughness classification based on DSM is limited. The surface roughness values calculated from the DSM picture the tree crowns, which are classified as rough. But the crowns usually don't interact with an avalanche flow (except powder snow avalanches). Therefore, within dense forests, DTM should be applied to calculated the surface roughness and DSM should only be applied for open areas, where roughness may still interact with the hazard process, but not being included in the forest classification. In this way areas with increased roughness outside of defined forest may be detected and included within the hazard modelling. In the case of avalanches, the RAMMS simulation tool (Christen et al., 2010) offers a possibility to add an area with increased friction parameters. A smart combination of DSM and DTM data may allow for better estimation of the surface roughness faced by the gravitational mass movement."

> 5- *The author mentioned that a relatively low-resolution DSM (1 m) can achieve better surface roughness (although I don't know how the author judged it). if the direction did so, has the author tested other lower-resolution data? For example, a global scale of 15m, 30m, etc.*

We did not test low-resolution data on global scale, since coarse resolutions do not capture small-scale surface roughness (Vanderhoof and Burt, 2018). Such roughness might be extremely important for the frequent avalanches, delineating the release areas (Veitinger et al., 2016).

In the section 1 Introduction we also mention the importance of high-resolution data for distinguishing more detailed terrain: "Higher DEM resolutions (< 1 m) allow us to see more detailed terrain, but they are usually only available for smaller areas."

The resolution of 1 m performed overall better than the other studied resolutions (0.1 and 0.5 m) for all studied algorithms. For our analysis was important that the surface roughness algorithm could distinguish well between the selected roughness categories and we used the paired Wilcoxon test to detect the overlapping distribution of pairs. We determined as well the point of minimum overlap, which we proposed to be used as a threshold for distinguishing between these roughness categories using the vector ruggedness measure algorithm.

**Response to the Community Comment #1**

Dear Sebastiano Trevisani,

Thank you for your comments to our paper. Regarding the points you highlighted we addressed and discussed them below.

- *A first point is related to flow directional roughness, as expressed in these two sentences:*

*Lines 103-104 : "Some attempts to calculate the roughness along a given direction have been made, but they have not yet been applied to large-scale hazard mapping (Michelini, 2016; Trevisani and Cavalli, 2016)."*

*Lines 108-109: " (3) Is it possible to improve the roughness calculation by introducing a directional roughness along the predominant mass flow direction?"*

*Being one of the authors of the cited paper (i.e., Trevisani and Cavalli, 2016) and having worked directly to the implementation of the surface roughness algorithms, as for example the indices based on the median of absolute directional differences (MAD, Trevisani and Rocca, 2015), I feel useful to furnish some more information to the readers.*

*The algorithm of flow-directional roughness presented in Trevisani and Cavalli 2016 (https://esurf.copernicus.org/articles/4/343/2016/) is fully working, even if a prototype, and can be applied at any scale, any resolution and for any given task, including natural hazards. In the cited study, for example, the algorithm has been applied to an area of 500 km2 with a LiDAR derived DTM (2 m pixel size); in general, there are no limitations for the size of DEM (i.e., it depends on available computational resources and to the given implementation). In that paper, we applied it as a coefficient in the sediment connectivity evaluation (Cavalli et al., 2013) and as a geomorphic tool for distinguishing morphologies using differences between isotropic and flow-directional roughness (e.g., for individuating gullies, landslide scarps, etc.). Consequently, to your question "Is it possible to improve the roughness calculation by introducing a directional roughness along the predominant mass flow direction?" the reply is "yes", as demonstrated in the cited paper. Clearly, that one is not the only possible approach and could be improved, but it seems to work well and put the basis for further developments.*

*The algorithm used in the paper is based on the MAD algorithm implemented as ArcMap tool and available in Github (see Trevisani and Rocca, 2015). The implementation is straightforward and can be implemented in other environments working with kernel based approaches for image analysis (e.g., Envi, raster package in R, Surfer, google earth engine, etc.).*

Thanks to your comment and after a literature review involving the directional roughness we will rephrase the following sentence

Lines 103-104 " **Some attempts to calculate the roughness along a given direction have been made, but they have not yet**

**been applied to large-scale hazard mapping (Michelini, 2016;  Trevisani and Cavalli, 2016).**"

"**However, the investigated natural hazards have a predominant diffusion direction identified as the combination of terrain slope and curvature. Some studies implemented the surface roughness along a predefined direction (Michelini, 2016; Trevisani and Rocca, 2015). The direction for which roughness has been computed, usually derived through GIS algorithm (D8 or D-infinity), applied to the original or smoothed elevation models. However, the direction derived through neighbourhood cells analysis is not always equal to the direction of the mass flow propagation. Such behaviours are sometimes observed when the routing volumes are extreme and therefore in some particular situations the propagation direction may be defined by its inertia rather than the topography (Guo et al., 2020). In other cases, the particular mountain topography may force mass flows to affect the opposite hillside of the valley through a runup mechanism (Iverson et al., 2016). Furthermore, the flow direction of banks and channel sides features computed with GIS algorithms do not usually correspond to the mass flow direction. In this situation bank direction can be improved through a smoothing process of the DTM in order to remove gullies and channel from the basal topography. This technique can be easily applicable in case of regular channels but it could become more complex when the channel morphology is irregular, since it could oversimplify the basal topography. For such reasons in this study, we propose a novel approach to calculate surface roughness along user defined lines.**"

As reported in the method section (lines 264-274) we developed a new directional roughness algorithm where the flow direction is established by the user through geospatial polylines. We then calculated the roughness as the standard deviation of the residual topography for the cells identified by the flow direction in the 3x3 moving window. As reported in the sentences above the manually identified flow directions can be more reliable with respect to the direction derived from a neighbourhood cell analysis, in case of particular routing mass flow behaviours. However, the flow direction can also be computed through a GIS algorithm and then used in the directional roughness algorithm reported in our study. Furthermore, we increased the number of directional roughness computation to 16 in comparison to the commonly used eight directions (D8 algorithm). For the technical implementation of the algorithm, we refer to the script available at the following link: https://github.com/TommBagg/terrain_roughness_GRASS.

- *The second point is related to the calculation of roughness indices e.g. line 107:*

*"(1) How well can different surface roughness categories be distinguished with the selected algorithms?"*

*I think that in the paper it could be important to highlight further that you tested standard approaches for roughness calculation and others have not been considered. For example, the MAD algorithm (applicable both for roughness calculation as well as for image analysis) has been developed with 3 main ideas in mind. The first, as a development from variogram based approaches (and with analogies to Local binary pattern and gray-level co-occurrence matrices), is to accept the fact that the surface roughness (or the synonym: surface texture) is a complex entity and that there are multiple aspects of surface roughness/texture that can be computed at multiple scales (e.g. anisotropy, short-range roughness, relative roughness, etc.). The second was to overcome some of the issues inherent to roughness measures such as the variogram and the standard elevation of DEM derivatives (residual topography, slope, etc..) that are affected by nonstationarity in data and by the presence of outliers. The third, was to create indices easy to interpret according to studied processes. Accordingly, as reported in Trevisani and Rocca 2015, even the short-range isotropic roughness calculated with MAD works much better than the standard deviation of residual relief or the one estimated with a variogram-based approach. Finally, the basic idea of that algorithm could move on further with the development of more complex approaches such as the capability to adapt locally to the "wavelengths" of morphologies (with similar approaches to Lindsay et al., 2019) or*

*detecting curvilinear structures via multipoint statistical indices (e.g., Mariethoz, G. & Lefebvre 2014).*

In our study, we considered the most commonly used roughness indices for hazard modelling. We selected the roughness derived through standard deviation and vector dispersion approaches applied in a certain moving window. In this way, the roughness algorithms we analysed can be directly applied to available elevation models.

We further integrate the paragraph involving the roughness algorithm selection. We therefore modified the 2.2 Surface roughness algorithms.

"**In order to describe the roughness, which consists of both geomorphological features and vegetation, we selected and tested seven algorithms using high-resolution DSMs. We selected widely used roughness algorithms already applied in the context of natural hazard modelling (Bühler et al., 2013; Crosta and Agliardi, 2004; Pfeiffer and Bowen, 1989; Veitinger and Sovilla, 2016; Wang and Lee, 2010). They are based on standard deviation and vector dispersion approaches calculated in a certain moving window. We then tested them with different spatial resolutions (0.1 m, 0.5 m and 1 m) and moving window areas (9 m$^{-2}$, 25 m$^{-2}$ and 49 m$^{-2}$) on both study areas. The selected algorithms are summarized in Table 1.**"

- *In the paragraph "This approach is widely used because it can be applied to different data types, such as point clouds (Vetter et al., 2012), satellite imagery (Gille et al., 2000; Schumann et al., 2007) and DEMs (Glenn et al., 2006; Trevisani and Cavalli, 2016)." I suggest changing "Trevisani and Cavalli, 2016" with "Cavalli and Marchi, 2008", because it is one of the first applications of standard deviation of residual DTM to LiDAR-based DTMs.*

Thanks for this observation. We will change the reference in line 204 from "Trevisani and Cavalli, 2016" to "**Cavalli and Marchi, 2008**" as you suggested.

**References**

Busse, A. and Jelly, T. O.: Influence of Surface Anisotropy on Turbulent Flow Over Irregular Roughness, Flow, Turbul. Combust., 104(2–3), 331–354, doi:10.1007/s10494-019-00074-4, 2020.

Bühler, Y., Adams, M. S., Bosch, R. and Stoffel, A.: Mapping snow depth in alpine terrain with unmanned aerial systems (UASs): Potential and limitations, Cryosphere, 10(3), 1075–1088, doi:10.5194/tc-10-1075-2016, 2016.

Bühler, Y., Kumar, S., Veitinger, J., Christen, M., Stoffel, A. and Snehmani: Automated identification of potential snow avalanche release areas based on digital elevation models, Nat. Hazards Earth Syst. Sci., 13(5), 1321–1335, doi:10.5194/nhess-13-1321-2013, 2013.

Cavalli, M. and Marchi, L.: Characterisation of the surface morphology of an alpine alluvial fan using airborne LiDAR, Nat. Hazards Earth Syst. Sci., 8, 323–333, doi:10.5194/nhess-8-323-2008, 2008.

Christen, M., Kowalski, J. and Bartelt, P.: RAMMS: Numerical simulation of dense snow avalanches in three-dimensional terrain, Cold Reg. Sci. Technol., 63(1–2), 1–14, doi:10.1016/j.coldregions.2010.04.005, 2010.

Crosta, G. B. and Agliardi, F.: Parametric evaluation of 3D dispersion of rockfall trajectories., 2004.

Deng, Y., Wilson, J. P. and Bauer, B. O.: DEM resolution dependencies of terrain attributes across a landscape, Int. J. Geogr. Inf. Sci., 21(2), 187–213, doi:10.1080/13658810600894364, 2007.

Frank, F., McArdell, B. W., Oggier, N., Baer, P., Christen, M. and Vieli, A.: Debris-flow modeling at

Meretschibach and Bondasca catchments, Switzerland: Sensitivity testing of field-data-based entrainment model, Nat. Hazards Earth Syst. Sci., 17(5), 801–815, doi:10.5194/nhess-17-801-2017, 2017.

Glenn, N. F., Streutker, D. R., Chadwick, D. J., Thackray, G. D. and Dorsch, S. J.: Analysis of LiDAR-derived topographic information for characterizing and differentiating landslide morphology and activity, Geomorphology, 73(1–2), 131–148, doi:10.1016/j.geomorph.2005.07.006, 2006.

Grohmann, C. H., Smith, M. J. and Riccomini, C.: Multiscale analysis of topographic surface roughness in the Midland Valley, Scotland, IEEE Trans. Geosci. Remote Sens., 49(4), 1200–1213, doi:10.1109/TGRS.2010.2053546, 2011.

Guo, J., Yi, S., Yin, Y., Cui, Y., Qin, M., Li, T. and Wang, C.: The effect of topography on landslide kinematics: a case study of the Jichang town landslide in Guizhou, China, Landslides, 17(4), 959–973, doi:10.1007/s10346-019-01339-9, 2020.

Habtezion, N., Tahmasebi Nasab, M. and Chu, X.: How does DEM resolution affect microtopographic characteristics, hydrologic connectivity, and modelling of hydrologic processes?, Hydrol. Process., 30(25), 4870–4892, doi:10.1002/hyp.10967, 2016.

Hungr, O. and McDougall, S.: Two numerical models for landslide dynamic analysis, Comput. Geosci., 35(5), 978–992, doi:10.1016/j.cageo.2007.12.003, 2009.

Hutchinson, M. F.: A locally adaptive approach to the interpolation of digital elevation models, in Proceedings of the Third International Conference/Workshop on Integrating GIS and Environmental Modelling, National Centre for Geographic Information and Analysis, Santa Barbara, CA, USA., 1996.

Insua-Arévalo, J. M., Tsige, M., Sánchez-Roldán, J. L., Rodríguez-Escudero, E. and Martínez-Díaz, J. J.: Influence of the microstructure and roughness of weakness planes on the strength anisotropy of a foliated clay-rich fault gouge, Eng. Geol., 289, 106186, doi:10.1016/j.enggeo.2021.106186, 2021.

Iverson, R. M., George, D. L. and Logan, M.: Debris flow runup on vertical barriers and adverse slopes, J. Geophys. Res. Earth Surf., 121(12), 2333–2357, doi:10.1002/2016JF003933, 2016.

Keijsers, J. G. S., Schoorl, J. M., Chang, K. T., Chiang, S. H., Claessens, L. and Veldkamp, A.: Calibration and resolution effects on model performance for predicting shallow landslide locations in Taiwan, Geomorphology, 133(3–4), 168–177, doi:10.1016/j.geomorph.2011.03.020, 2011.

Koponen, P., Nygren, P., Sabatier, D., Rousteau, A. and Saur, E.: Tree species diversity and forest structure in relation to microtopography in a tropical freshwater swamp forest in French Guiana, Plant Ecol., 173(1), 17–32, doi:10.1023/B:VEGE.0000026328.98628.b8, 2004.

Mergili, M., Fischer, J. T., Krenn, J. and Pudasaini, S. P.: R.avaflow v1, an advanced open-source computational framework for the propagation and interaction of two-phase mass flows, Geosci. Model Dev., 10(2), 553–569, doi:10.5194/gmd-10-553-2017, 2017.

Michelini, T.: Analisi sperimentale delle scabrezze di superficie e di fondo per la modellazione dinamica dei flussi torrentizi e della caduta massi. [online] Available from: http://paduaresearch.cab.unipd.it/9407/1/Tesi_Tamara_Michelini.pdf, 2016.

Middleton, M., Nevalainen, P., Hyvönen, E., Heikkonen, J. and Sutinen, R.: Pattern recognition of LiDAR data and sediment anisotropy advocate a polygenetic subglacial mass-flow origin for the Kemijärvi hummocky moraine field in northern Finland, Geomorphology, 362, 107212, doi:10.1016/j.geomorph.2020.107212, 2020.

Pastore, M. and Calcagnì, A.: Measuring distribution similarities between samples: A distribution-free overlapping index, Front. Psychol., 10(May), 1–8, doi:10.3389/fpsyg.2019.01089, 2019.

Pastore, M.: Overlapping: a R package for estimating overlapping in empirical distributions, J. Open Source Softw., 3(32), 1023, doi:10.21105/joss.01023, 2018.

Pfeiffer, T. J. and Bowen, T. D.: Computer simulation of rockfalls, Bull. Int. Assoc. Eng. Geol., 26(1), 135–146, doi:10.2113/gseegeosci.xxvi.1.135, 1989.

Pudasaini, S. P. and Mergili, M.: A Multi-Phase Mass Flow Model, J. Geophys. Res. Earth Surf., 124, 1–23, doi:10.1029/2019jf005204, 2019.

R Core Team: R: A language and environment for statistical computing, R Found. Stat. Comput. [online] Available from: http://www.r-project.org (Accessed 28 January 2021), 2021.

Rosatti, G. and Begnudelli, L.: Two-dimensional simulation of debris flows over mobile bed: Enhancing the TRENT2D model by using a well-balanced Generalized Roe-type solver, Comput. Fluids, 71, 179–195, doi:10.1016/j.compfluid.2012.10.006, 2013.

Roy, S. G., Koons, P. O., Osti, B., Upton, P. and Tucker, G. E.: Multi-scale characterization of topographic anisotropy, Comput. Geosci., 90, 102–116, doi:10.1016/j.cageo.2015.09.023, 2016.

Sappington, J. M., Longshore, K. M. and Thompson, D. B.: Quantifying Landscape Ruggedness for Animal Habitat Analysis: A Case Study Using Bighorn Sheep in the Mojave Desert, J. Wildl. Manage., 71(5), 1419–1426, doi:10.2193/2005-723, 2007.

Tarolli, P. and Tarboton, D. G.: Hydrology and Earth System Sciences A new method for determination of most likely landslide initiation points and the evaluation of digital terrain model scale in terrain stability mapping. [online] Available from: www.hydrol-earth-syst-sci.net/10/663/2006/ (Accessed 28 June 2021), 2006.

Trevisani, S. and Rocca, M.: MAD: Robust image texture analysis for applications in high resolution geomorphometry, Comput. Geosci., 81, 78–92, doi:10.1016/j.cageo.2015.04.003, 2015.

Trevisani, S. and Cavalli, M.: Topography-based flow-directional roughness: potential and challenges, Earth Surf. Dyn., 4(2), 343–358, doi:10.5194/esurf-4-343-2016, 2016.

Vanderhoof, M.K.; Burt, C. Applying High-Resolution Imagery to Evaluate Restoration-Induced Changes in Stream Condition, Missouri River Headwaters Basin, Montana. Remote Sens., 10, 913. https://doi.org/10.3390/rs10060913, 2018.

Viero, D. Pietro and Valipour, M.: Modeling anisotropy in free-surface overland and shallow inundation flows, Adv. Water Resour., 104, 1–14, doi:10.1016/j.advwatres.2017.03.007, 2017.

Veitinger, J., Purves, R. S., and Sovilla, B.: Potential slab avalanche release area identification from estimated winter terrain: a multi-scale, fuzzy logic approach. Natural Hazards and Earth System Science, 16(10), 2211-2225. https://doi.org/10.5194/nhess-16-2211-2016, 2016.

Viero, D. Pietro and Valipour, M.: Modeling anisotropy in free-surface overland and shallow inundation flows, Adv. Water Resour., 104, 1–14, doi:10.1016/j.advwatres.2017.03.007, 2017

Wang, I. T. and Lee, C. Y.: Influence of slope shape and surface roughness on the moving paths of a single rockfall, World Acad. Sci. Eng. Technol., 65(5), 1021–1027, doi:10.5281/zenodo.1059436, 2010.

Wu, J., Yang, Q. and Li, Y.: Partitioning of terrain features based on roughness, Remote Sens., 10(12), 1–21, doi:10.3390/rs10121985, 2018.

---

## Author Response (AR2)

Dear Referee #2,

Thank you very much for your comments, we changed the text accordingly and we adjusted the units and labels at all figures.

*- Line 22 RAMMS, should write the full name at first appearance.*

Thank you for this remark. We added the full name (see in bold).

We simulated avalanches on a different elevation models (LiDAR-based) to observe a potential influence of a DSM and a digital terrain model (DTM) using simulation tool **Rapid Mass Movement Simulation** (RAMMS).

*- Line 34 something went wrong here*

*….and ecosystem services (; Koponen et al., 2004;;;; Wu et al., 2018).*

Quantifying surface roughness is thus central for the estimation of various biophysical characteristics and ecosystem services (Koponen et al., 2004; Wu et al., 2018).

*- Figure 4 you used the format as m2, but you also used the format as m-2 in other places, please be unified.*

Thank you for this observation, we changed the format to $m^2$ throughout the manuscript (including all the figures) to refer for the moving window area.

*- Figure 4 the labels of the y-axis are impossible to read.*

We increased the size of the labels.

*- Figure 5 why you changed the label of the sub-picture from English to number?*

We have not changed Figure 5 from the first submission. The labels are numbers for easier reference and have the name in English above each sub-picture as well.